# Semi-supervised Learning of Partial Differential Operators and Dynamical Flows

## Abstract

The evolution of many dynamical systems is generically governed by nonlinear partial differential equations (PDEs), whose solution, in a simulation framework, requires vast amounts of computational resources. In this work, we present a novel method that combines a hyper-network solver with a Fourier Neural Operator architecture. Our method treats time and space separately and as a result, it successfully propagates initial conditions in continuous time steps by employing the general composition properties of the partial differential operators. Following previous works, supervision is provided at a specific time point. We test our method on various time evolution PDEs, including nonlinear fluid flows in one, two, or three spatial dimensions. The results show that the new method improves the learning accuracy at the time of the supervision point, and can interpolate the solutions to any intermediate time.

## 1 Introduction

The evolution of classical and quantum physical dynamical systems in space and time is generically modeled by non-linear partial differential equations. Such are, for instance, Einstein equations of General Relativity, Maxwell equations of Electromagnetism, Schrödinger equation of Quantum Mechanics and Navier-Stokes (NS) equations of fluid flows. These equations, together with appropriate initial and boundary conditions, provide a complete quantitative description of the physical world within their regime of validity. Since these dynamic evolution settings are governed by partial differential operators that are often highly non-linear, it is rare to have analytical solutions for dynamic systems. This is especially true when the system contains a large number of interacting degrees of freedom in the non-linear regime.

Consider, as an example, the NS equations, which describe the motion of viscous fluids. In the regime of high Reynolds numbers of the order of one thousand, one observes turbulences, in which all symmetries are broken. Despite much effort, two basic questions have remained unanswered, that is the existence and uniqueness of the solutions to the 3D NS equations and the anomalous scaling of the fluid observables in statistical turbulence. The solution to the (deterministic) NS equations seems almost random and is very sensitive to the initial conditions. Many numerical techniques have been developed for constructing and analysing the solutions to fluid dynamics systems. However, the complexity of these solvers grows quickly as the spacing in the grid that is used for approximating the solution is reduced and the degrees of freedom of the interacting fluid increases.

Given the theoretical and practical importance of constructing solutions to these equations, it is natural to ask whether neural networks can learn such evolution equations and construct new solutions. The two fundamental questions are: (i) The ability to generalize to initial conditions that are different from those presented in the training set, and (ii) The ability to generalize to unseen time points, not provided during training. The reason to hope that such tasks can be performed by machine learning is that despite the seemingly random behaviour of, e.g. fluid flows in the turbulent regime, there is an underlying low-entropy structure that can be learnt. Indeed, in diverse cases, neural network-based solvers have been shown to provide comparable results to other numerical methods, while utilizing fewer resources.

**Our Contributions** We present a hyper-network based solver combined with a Fourier Neural Operator architecture which is able to learn non-linear partial differential operators that govern the dynamics of chaotic and out of the equilibrium flows.

1. Our hyper-network architecture treats time and space separately. Utilizing a data set of initial conditions and the corresponding solutions at a labeled fixed time, the network learns a large class of time evolution PDEs.

2. Our approach enables interpolation to arbitrary (unlabelled) continuous times without additional data points.

3. Our solutions improve the learning accuracy at the supervision time-points.

4. We thoroughly test our method on various time evolution PDEs, including non-linear fluid flows in one, two and three spatial dimensions.

## 2 RELATED WORK

**Hyper-networks** While conventional networks employ a fixed set of pre-determined parameters, which is independent of the input, the hyper-network scheme, invented multiple times, and coined by Ha et al. (2017), allows the parameters of a neural network to explicitly rely on the input by combining two neural networks. The first neural network, called the hyper-network, processes the input or part of it and outputs the weights of a second neural network. The second network, called the primary network, has a fixed architecture and weights that vary based on the input. It returns, given its input, the final output.

This framework was used successfully in a variety of tasks, ranging from computer vision (Littwin & Wolf, 2019), continual learning (von Oswald et al., 2020), and language modeling (Suarez, 2017). While it is natural to learn functions with hyper-networks, since the primary network can be seen as a dynamic, input-dependent function, we are not aware of any previous work that applies this scheme for recovering physical operators.

**Neural network-based PDE solvers** Due to the well-known limitations of traditional PDE solvers on one hand, and in light of new advances made in the field of neural networks on the other, lately we have witnessed very significant progress in the field of neural network-based PDE solvers (Karniadakis et al., 2021). These solvers can be roughly divided into two groups according to the resource they utilize for learning: data-driven and model-based.

Model-based solvers, known as Physics Informed Neural Networks (PINNs) (Raissi et al., 2019), harness the differential operator itself for supervision. This is done by defining a loss, the residual of the PDE. These solvers also require a training dataset and can provide solutions for arbitrary times.

Data-driven solvers are trained over a large dataset containing initial conditions and observed final states after the same time interval, $T$. Among these solvers there are encoder-decoder based ones, either a fully convolutional (Zhu & Zabaras, 2018) one, or a U-Net shaped one (Thuerey et al., 2020), the PCANN (Bhattacharya et al., 2021) which applies Principal Component Analysis (PCA) as an auto-encoder and interpolates between the latent spaces using a neural network, The Multipole Graph Neural Operator (MGNO), the DeepONet which encodes input function and locations separately using two neural networks (Lu et al., 2019b), Graph Neural Simulators Sanchez-Gonzalez et al. (2020) which utilize graph neural networks, and the MP-PDE (Brandstetter et al., 2021), which learns a neural operator on the graph . These solvers learn solutions on a specific discretized grid, which poses a limitation for any practical applications.

Recently, a mesh-invariant data-driven direction has been proposed (Lu et al., 2019a; Nelsen & Stuart, 2021; Anandkumar et al., 2020; Patel et al., 2021). The mesh invariance is obtained by learning operators rather than mappings between initial and final states. This is achieved using architectures that enforce mesh invariant network parameters. Such networks have the ability train and infer on different meshes. As a result, these networks can be trained on small grids and run, during inference, on very large grids. Li et al. (2020b) have advanced the mesh-invariant line of works by introducing Fourier Neural Operators (FNO). FNOs utilize both a convolution layer in real space and a Fourier layer in the Fourier domain. Each Fourier layer first transforms its input, $x \in \mathbb{R}^d$ to Fourier space, point-wise multiples the transformed input by a set of learned coefficients, $w \in \mathbb{R}^{\frac{d}{2}}$ (note that since $x$ is a real vector, only $\frac{d}{2}$ coefficients are required), and applies an inverse Fourier transform to obtain the output, $y = \mathcal{F}^{-1}\{w \odot \mathcal{F}\{x\}\} \in \mathbb{R}^d$. Since the learned parameters reside in Fourier space, it allows training on a specific scale, and inferring on other scales. It has been shown that the FNO solver outperforms previous solvers in a number of important PDEs. The current data-driven methods

successfully evolve a solution to the supervised time step. By successive applications of these methods, solutions may be evolved in increments of $T$. Though the accuracy of these solutions may degrade, some statistical properties can still be obtained (Li et al., 2021). However, these methods are not designed for evolving solutions to intermediate times. Knowing the solutions along the complete time evolution trajectory is highly valuable for theoretical and practical reasons and provides means for learning new dynamical principles.

## 3 PARTIAL DIFFERENTIAL EQUATIONS

Consider a $d$-dimensional vector field $\vec{v}(\vec{x}, t) : \mathbb{T}^d \times \mathbb{R} \to \mathbb{R}^d$, where $\vec{x} = (x_1, \ldots, x_d)$ are periodic spatial coordinates $x_i \simeq x_i + 2\pi$ on the d-dimensional torus, $\mathbb{T}^d$, and $t$ is the time coordinate. The vector field evolves dynamically from the initial condition $\vec{v}(\vec{x}, t = 0) : \mathbb{T}^d \to \mathbb{R}^d$ according to a non-linear PDE of the form,

$$\partial_t \vec{v}(\vec{x}, t) = \mathcal{L} \vec{v}(\vec{x}, t), \tag{1}$$

where $\mathcal{L}$ is a differential operator that does not depend explicitly on time, and has an expansion in $v$ and its spatial derivatives. We assume that given a regular bounded initial vector field there is a unique regular solution to equation 1. Since $\mathcal{L}$ does not depend explicitly on time, we can formally write the solution of such equations as

$$\vec{v}(\vec{x}, t) = e^{t\mathcal{L}} \vec{v}(\vec{x}, 0) \equiv \Phi_t \vec{v}(\vec{x}, 0). \tag{2}$$

Because of dissipation terms, such as the viscosity term in the NS equations, solutions to equation 1 generically break time reversal invariance ($t \to -t, \vec{v} \to -\vec{v}$). Furthermore, the total energy of the system is non-increasing as a function of time since we are not injecting energy at $t \neq 0$, $\partial_t \int d^d x \frac{\vec{v} \cdot \vec{v}}{2} \leq 0$, and serves as a consistency on the network solutions.

In this work, we consider the following non-linear PDEs:

**Generalized Burgers equation**. Describes one-dimensional compressible fluid flows with scalar velocity field $v(x, t)$. We use the generalization parameter with $q = 1, 2, 3, 4$ ($q = 1$ case corresponds to the regular Burgers equation),

$$\partial_t v + v^q \nabla v = \nu \Delta v, \tag{3}$$

where $\nu$ is the kinematic viscosity, $\nabla$ the gradient and $\Delta$ is the Laplacian

**Chafee–Infante equation**. One-dimensional equation with a constant $\lambda$ parameter that models reaction-diffusion dynamics,

$$\partial_t v + \lambda(v^3 - v) = \Delta v. \tag{4}$$

**Two-dimensional Burgers**. Describes compressible fluid flows in two space dimensions for the vector field, $\vec{v}(\vec{x}, t) = (v^1, v^2)$,

$$\partial_t \vec{v} + (\vec{v} \cdot \nabla)\vec{v} = \nu \Delta \vec{v}. \tag{5}$$

**Two and three-dimensional NS**. Two and three-dimensional incompressible NS equations,

$$\partial_t \vec{v} + (\vec{v} \cdot \nabla)\vec{v} = -\nabla p + \nu \Delta \vec{v}, \quad \nabla \cdot \vec{v} = 0, \tag{6}$$

where $p$ is the fluid's pressure, $\vec{v}(\vec{x}, t) = (v^1, \ldots, v^d)$ and $\nu$ is the kinematic viscosity.

## 4 METHOD

Typically, a data-driven PDE solver, such as a neural network, evolves unseen velocity fields, $\vec{v}(\vec{x}, t = 0)$ to a fixed time $T$,

$$\Phi_T \vec{v}(\vec{x}, t = 0) = \vec{v}(\vec{x}, T), \tag{7}$$

by learning from a set of $i = 1 \ldots N$ initial conditions sampled at $t = 0$, $\vec{v}_i(\vec{x}, t = 0)$, and their corresponding time-evolved solutions of $\vec{v}_i(\vec{x}, t = T)$ of equation 1.

We generalize $\Phi_T$, to propagate solutions at intermediate times, $0 \leq t \leq T$, by elevating a standard neural architecture to a hyper-network one. A hyper-network architecture is a composition of two neural networks, a primary network $f_w(x)$ and a hypernetwork $g_\theta(t)$ with learned parameters $\theta$, such

that the parameters $w$ of $f$ are given as the output of $g$. Unlike common neural architectures, where a single input is mapped to a single output, in this construction, the input $t$ is mapped to a function, $f_{g_\theta(t)}$, which maps $x$ to its output. Applying this to our case we have,

$$\Phi_t \vec{v}(\vec{x}, 0) = f_{g_\theta(t)}(\vec{v}(\vec{x}, 0)) = \vec{v}(\vec{x}, t). \tag{8}$$

This architecture may be used to learn not only the time-evolved solutions at $t = T$, but also intermediate solutions, at $0 < t < T$, without explicitly providing the network with any intermediate time solution. This task may be accomplished by utilizing general consistency conditions, which apply to any equation of the form of equation 1,

$$\Phi_{t_1} \Phi_{t_2} = \Phi_{t_2} \Phi_{t_1} = \Phi_{t_1 + t_2} . \tag{9}$$

Notice that $\Phi_0 = \mathbb{I}_d$ follows from equation 9. Consider the differential equation 1. Suppose we construct a map $\Phi_t$ such that $\Phi_T \vec{v}_i (\vec{x}, t = 0) = \vec{v}_i (\vec{x}, t = T)$, where $\vec{v}_i (\vec{x}, t = 0)$ are all initial conditions (or form an appropriate complete set that spans all possible initial conditions) for equation 1 and $\vec{v}_i (\vec{x}, t = T)$ are the corresponding solutions of the equation at time $T$. We would like to know whether $\Phi_t \vec{v}_i (\vec{x}, 0) = \vec{v}_i (\vec{x}, t)$ for all times $t \geq 0$, i.e. that the map $\Phi_t$ evolves all the initial condition correctly as the solutions of equation 1 for any time.

**Lemma 4.1.** *A map $\Phi_t$ that propagates a complete set of initial conditions at $t = 0$ to the corresponding solutions of equation 1 at a fixed time $t = T$ and satisfies the composition law in equation 9 propagates the initial conditions to the corresponding solutions for any time $t \geq 0$.*

*Proof.* Denote by $\varphi_t$ the map that propagates correctly at any time $t$ the initial conditions to solutions of equation 1. The space of initial conditions $\vec{v}(\vec{x}, t = 0) : \mathbb{T}^d \to \mathbb{R}^d$ is a complete set of functions and the same holds for $\vec{v}(\vec{x}, T)$. Since $\varphi_T$ agrees with $\Phi_T$ on this set of initial condition it follows that $\varphi_T \equiv \Phi_T$. Consider next a partition of the interval $[0, T]$ to $p$ identical parts, $[0, \frac{T}{p}], [\frac{T}{p}, \frac{2T}{p}], ..., [\frac{(p-1)T}{p}, T]$. We have the composition of maps:

$$\Phi_{\frac{T}{p}} \Phi_{\frac{T}{p}} \cdots \Phi_{\frac{T}{p}} \equiv \Phi_T \equiv \varphi_T \equiv \varphi_{\frac{T}{p}} \varphi_{\frac{T}{p}} \cdots \varphi_{\frac{T}{p}} , \tag{10}$$

from which we conclude that $\varphi_{\frac{T}{p}} \equiv \Phi_{\frac{T}{p}}$ (there can be an overall sign in the relation between them when $p$ is taken to be odd which is fixed to one by continuity). We can perform this division for any $p$ and conclude using the composition rule in equation 9 that $\Phi_t \equiv \varphi_t$ for any $t$. $\square$

## 4.1 Loss functions

In this section we introduce the loss functions used to train our models. In order to simplify our notation, we will suppress the explicit space dependence in our equations, so $\vec{v}(\vec{x}, t = 0)$ will be denoted by $\vec{v}(0)$. We denote by $N$ the size of the training set of solutions. Our loss function consists of several terms, supervised and unsupervised.

The information of the specific PDE is given to our model via a supervised term,

$$\mathcal{L}_{\text{final}} = \sum_{i=1}^{N} \text{Err} \left( \Phi_T \vec{v}_i (0) , \vec{v}_i (T) \right) , \tag{11}$$

which is responsible for propagating the initial conditions to their final state at $t = T$. The reconstruction error function Err is defined as,

$$\text{Err} \left( \vec{a}, \vec{b} \right) = \sqrt{\frac{\sum_\alpha \left( \vec{a}_\alpha - \vec{b}_\alpha \right)^2}{\sum_\beta \vec{b}_\beta^2}} , \tag{12}$$

where the $\alpha, \beta$ indices run over the grid coordinates.

To impose the consistency condition $\Phi_0 = \mathbb{I}_d$, we use the loss function term,

$$\mathcal{L}_{\text{initial}} = \sum_{i=1}^{N} \text{Err} \left( \Phi_0 \vec{v}_i (0) , \vec{v}_i (0) \right) . \tag{13}$$

The composition law in equation 9 is implemented by the loss function term,

$$\mathcal{L}_{\text{inter}} = \sum_{i=1}^{N} \text{Err}\left(\Phi_{t_1^i + t_2^i} \vec{v}_i(0), \Phi_{t_2^i} \Phi_{t_1^i} \vec{v}_i(0)\right), \tag{14}$$

where $\tilde{T} = t_1^i + t_2^i$ is drawn uniformly from $\mathbb{U}[0, T]$ and $t_i^1$ is drawn uniformly from $\mathbb{U}[0, \tilde{T}]$, so the sum of $t_1^i + t_2^i$ does not exceed $T$.

The last term concerns the composition of the maps, $\Phi_t$, to generate $\Phi_T$ by $p$-sub-intervals of the interval $[0, T]$,

$$\prod_{j=1}^{p} \Phi_{t_j} = \Phi_T, \quad \sum_{j=1}^{p} t_j = T, \quad t_j > 0, \tag{15}$$

and it reads,

$$\mathcal{L}_{\text{comp}}^{(P)} = \sum_{i=1}^{N} \sum_{p=2}^{P} \text{Err}\left(\prod_{j=1}^{p} \Phi_{t_j} \vec{v}_i(0), \vec{v}_i(T)\right), \tag{16}$$

where $P$ is the maximal number of intervals used. The time intervals are sampled using $t_j \sim \mathbb{U}\left[\frac{j-1}{p}T, \frac{j}{p}T\right]$, and the last interval is dictated by the constraint in equation 15. The $P = 1$ term is omitted, since it corresponds to $\mathcal{L}_{\text{final}}$. The total loss function reads,

$$\mathcal{L}_{\text{tot}}^{(P)} = \mathcal{L}_{\text{final}} + \mathcal{L}_{\text{initial}} + \mathcal{L}_{\text{inter}} + \mathcal{L}_{\text{comp}}^{(P)}. \tag{17}$$

## 5 Experiments

We present a battery of experiments in 1D, 2D and 3D. The main baseline we use for comparison is FNO (Anandkumar et al., 2020), which is the current state of the art and shares the same architecture as our primary network $f$. For the 1D Burgers equations, we also compare with the results of Multipole Graph Neural Operator (MGNO) (Li et al., 2020a). We do not have the results of this method for other datasets, due to its high runtime and memory complexity.

### 5.1 Experimental Setup

Our neural operator is represented by a hyper-network composition of two networks. For the primary network $f$, we use the original FNO architecture. This architecture consists of four Fourier integral operators followed by a ReLU activation function. The width of the Fourier integral operators is 64, 32 and 20, and the operators integrate up to a cutoff of 16, 12, 4 modes in Fourier space, for one, two, and three dimensions, respectively. Recently, Kovachki et al. (2021) demonstrated that the FNO architecture may benefit from replacing the ReLU activation with a GeLU activation. Therefore, our results are reported using both variants, denoted by $+$ suffix for the GeLU variant.

For the hypernetwork $g$ we use a fully-connected network with three layers, each with a width of 32 units, and a ReLU activation function. $g$ generates the weights of network $f$ for a given time point $t$, thus mapping the scalar $t$ to the dimensionality of the parameter space of $f$. A $\tanh$ activation function is applied to the time dimension, $t$, before it is fed to $g$. Together, $f$ and $g$ constitute the operator $\Phi_t$ as appears in equation 8.

All experiments are trained for 500 epochs, with an ADAM Kingma & Ba (2015) optimizer and a learning rate of 0.001, a weight decay of 0.0001, and a batch size of 20. For MGNO, we used a learning rate of 0.00001, weight decay of 0.0005 and a batch size of 1, as in its official github repository. For our method, at each iteration and for each sample, we sample different time points for obtaining the time intervals of $\mathcal{L}_{\text{comp}}^{(P)}$ and $\mathcal{L}_{\text{inter}}$ as detailed in Section 4.1.

For the one-dimensional PDEs, we randomly shift both the input and the output to leverage their periodic structure. For one- and two-dimensional problems, the four loss terms are equally weighted. For the three-dimensional NS equation, the $\mathcal{L}_{\text{inter}}$ term was multiplied by a factor of 0.1, the reason being that as we increase the number of space dimensions we encounter a complexity due to the growing number of degrees of freedom (i.e. the minimum number of grid points per integral scale)

required to accurately describe a fluid flow. Standard Kolmogorov's type scaling (Frisch, 1995) implies that the number of degrees of freedom needed scales as $N_d \sim \mathcal{R}^{\frac{3d}{4}}$, where $\mathcal{R}$ is the Reynolds number. This complexity is seen in numerical simulations and we observe it in our framework of learning, as well. All experiments were executed using an NVIDIA RTX 2080 TI with 24GB.

## 5.2 DATA PREPARATION

Our experiments require a set of initial conditions at $t = 0$ and time-evolved solutions at time $t = T$ for our PDEs. For the simplicity of notations we will set $T = 1$ in the following description. In all cases, the boundary conditions as well as the time-evolved solutions are periodic in all spatial dimensions. The creation of these datasets consists of two steps. The first step is to sample the initial conditions from some distribution. For all of our one-dimensional experiments we use the same initial conditions from the dataset of Li et al. (2020b), which was originally used for Burgers equations and sampled using Gaussian random fields. For Burgers equation, we used solutions sampled at $512, 1024, 2048, 4096, 8192$, while for the other PDEs we used a grid size of $1024$. All one-dimensional PDEs assume a viscosity, $\nu = 0.1$.

For the two-dimensional Burgers equation our data is sampled using a Gaussian process with a periodic kernel,

$$k(x, y; x', y') = \sigma^2 e^{-\frac{2 \sin^2(|x-x'|/2)}{l^2}} e^{-\frac{2 \sin^2(|y-y'|/2)}{l^2}}, \tag{18}$$

with $\sigma = 1$ and $l = 0.6$, and with a grid size $64 \times 64$ and a viscosity of $\nu = 0.001$. For the two and three-dimensional NS equation we use the $\phi_{\text{Flow}}$ package (Holl et al., 2020) to generate samples on a $64 \times 64$ and $128 \times 128 \times 128$ grids, later resampled to $64 \times 64 \times 64$. For the two and three-dimensional NS, we use a viscosity of $\nu = 0.001$. In all cases the training and test sets consist of 1000 and 100 samples, respectively.

The second step is to compute the final state at $t = 1$. For the equations in one-dimension and the two-dimensional Burgers equations we use the `py-pde` python package (Zwicker, 2020)(MIT License) to evolve the initial state for the final solution. To evaluate the time evolution capabilities of our method, we compute all intermediate solutions with $\Delta t = 0.01$ time-step increments for the one-dimensional Burgers equation. For the two-dimensional Burgers equations we do this up to $t = 1.15$, since at this regime the solutions develop shock waves due to low viscosity and we learn regular solutions. For the two-dimensional and three-dimensional NS equations we use $\phi_{\text{Flow}}$ package (Holl et al., 2020)(MIT License) to increment the solutions with $\Delta t = 0.1$.

## 5.3 RESULTS

We compare the performance of our method on the one dimensional PDEs with FNO (Li et al., 2020b) and MGNO (Li et al., 2020a). In this scenario, we use a loss term with two intervals, $P = 2$. Since MGNO may only use a batch size of 1 and requires a long time to execute, we present it only for the one-dimensional Burgers equation. As seen in Table 1, in all PDEs, our method outperforms the original FNO by at least a factor of 3. The addition of a GeLU activation function, as suggested by Kovachki et al. (2021), is beneficial to FNO, but offers only a slight improvement to our method, and only on some of the benchmarks. We further compare the one-dimensional Burgers equation for various grid resolutions at $t = 1$ in Figure 1. As previously noted by Li et al. (2020b), the grid resolution does not influence the reconstruction error for any of the methods.

To assess the contribution of the $\mathcal{L}_{\text{inter}}$ term and to account for the influence of the number of intervals induced by $\mathcal{L}_{\text{comp}}^{(P)}$, we evaluate the reconstruction error for the task of interpolating in $t \in [0, 1]$ in comparison to the ground truth solutions in Figure 2. As a baseline for the interpolation capabilities of our network, we further apply a linear interpolation between the initial and final solution, $v(t) = (1 - t)v(0) + tv(1)$ (even though one uses the $t = 1$ solution explicitly for that which gives clear and unfair advantage at $t = 1$). The horizontal dashed cyan line marks the reconstruction error of the the FNO method at $t = 1$, thus the interpolation our approach provides exceeds the prediction ability of FNO at discrete time steps. For extrapolation, $t > 1$, we divide the time into unit intervals plus a residual interval (e.g. for $t = 2.3$ we have the three intervals $1, 1, 0.3$), and we let the network evolve the initial condition to the solution at $t = 1$, and then repeatedly use the solution as initial condition to estimate the solution at later times. As can be seen, the addition of the $\mathcal{L}_{\text{inter}}$ term dramatically improves the reconstruction error. Furthermore, increasing the number

Table 1: The reconstruction error on the one-dimensional PDEs at $t = 1$, grid size = 1024. B refers to Burgers equations, GB to generalized Burgers equations, and CI to Chafee–Infante equation.

| Method | B | CI | GB q=2 | GB q=3 | GB q=4 |
|---|---|---|---|---|---|
| MGNO | 0.0552 | - | - | - | - |
| FNO | 0.0096 | 0.0028 | 0.0090 | 0.0095 | 0.0098 |
| FNO+ | 0.0022 | 0.0010 | 0.0032 | **0.0025** | 0.0055 |
| Ours | 0.0022 | **0.0008** | 0.0031 | 0.0030 | **0.0030** |
| Ours+ | **0.0015** | 0.0011 | **0.0024** | 0.0026 | **0.0030** |

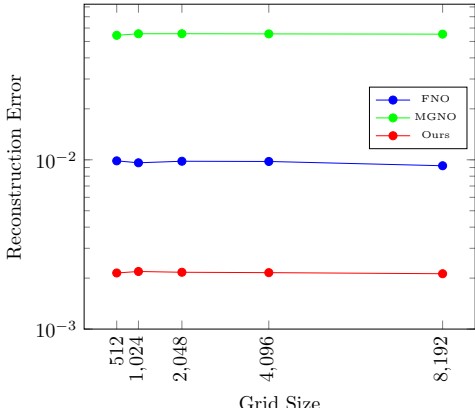

Figure 1: The reconstruction error on the one-dimensional Burgers equation for different grid resolutions.

of intervals $P$ induced by $\mathcal{L}_{\text{comp}}^{(P)}$ reduces the reconstruction error, but with a smaller effect. The reconstruction error in the extrapolation region is low mainly because of the high energy dissipation. Note that for all of the presented variations, the reconstruction error of our method at $t = 1$ is lower than FNO.

Table 2 contains a comparison of our method to FNO on the two dimensional PDEs, Burgers and NS equations. In two-dimensions the advantage of our method over the baseline at $t = 1$ is relatively modest. However, a much larger advantage is revealed on intermediate time, as can be seen in Figures 3a and in 3b. In both PDEs it is apparent that $\mathcal{L}_{\text{inter}}$ is an essential ingredient for providing a satisfactory interpolation, and that the required number of intervals, $P$, is not large. This is because $\mathcal{L}_{\text{inter}}$ enforces time-translation invariance, and therefore smooths the solution.

Table 3 summarizes the reconstruction error on the three-dimensional NS equation at $t = 1$. In this case, we applied our method with $P = 1$ and ReLU activations for computational reasons, as a larger number of intervals required significantly more memory (during training only; during inference it requires a memory size similar to that of FNO) and GeLU activation cannot be applied "in-place". As can be seen, our method outperforms FNO and performs the best with the $\mathcal{L}_{\text{inter}}$ term in this benchmark as well.

## 6 DISCUSSION AND OPEN QUESTIONS

Lemma 1 points to practical sources of possible errors in learning the map $\Phi_t$. The first source of error is concerned with the training data set. Quantifying the set of initial conditions and solutions at time $T$, from which we can learn the map $\Phi_T$ with a required accuracy, depends on the complexity of the PDE. Burgers equation is integrable (for regular solutions) and can be analytically solved. In contrast, NS equations are not integrable dynamical flows and cannot be solved analytically. Thus, for a given learning accuracy, we expect the data set needed for Burgers equation to be much smaller than the one needed for NS equations. Indeed, we see numerically the difference in the learning accuracy

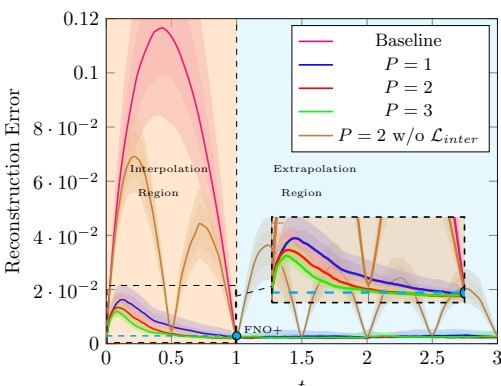

Figure 2: The reconstruction error on the one-dimensional Burgers equation for intermediate time predictions and the py-pde solver solutions for different numbers of intervals $P$ in the composition loss term, $\mathcal{L}_{\text{comp}}^{(P)}$. We show the median as well as the 10th, 25th, 75th and 90th percentiles. The statistics is based on 100 samples from the test set. The dashed horizontal cyan line marks the reconstruction error of FNO at $t = 1$ for comparison.

Table 2: The reconstruction error on the two-dimensional PDEs at $t = 1$, Burgers and NS equations.

| Method | 2D Burgers | 2D NS |
|--------|-----------|--------|
| FNO    | 0.0336    | 0.4340 |
| FNO+   | 0.0248    | 0.4511 |
| Ours   | 0.0335    | 0.4302 |
| Ours+  | **0.0214** | **0.4163** |

of these PDEs. Second, there is error in the learning of the consistency conditions in equation 9. Third, we impose only a limited set of partitions in equation 15, which leads to an error that scales as $\frac{1}{P}$, where $P$ is the partition number of the time interval. We decreased this error by applying various such partitions. Note, that the non-convexity of the loss minimization problem implies, as we observe in the network performance, that the second source of error tends to increase when decreasing the third one and vice versa. The difference in the complexity of the NS equations compared to Burgers equation is reflected also in the interpolation performance, which is significantly better for the Burgers flows in Figure 3a compared to the NS flows in Figure 3b.

Our work opens up several directions for further interesting studies. First, it is straightforward to generalize our framework to other PDEs, such as having higher-order time derivatives, as in relativistic systems where the time derivative is of order two.

Second, we only considered regular solutions in our work. It is known that singular solutions such as shock waves in compressible fluids play an important role in the system's dynamics. The continuation of the solutions passed the singularity is known to be unique in one-dimensional Burgers system. This is not the case in general as, for instance, for inviscid fluid flows modelled by the incompressible Euler equations. It would be of interest to study how well our network learns such solutions and extends them beyond the singularity.

Third, we learnt the dynamical equations at a fixed grid size of the spatial coordinates. It would be highly valuable if the network could learn to interpolate to smaller grid sizes. FNO allows a super-resolution and hence an interpolation to smaller grid size. It does so by employing the same physics of the larger scale. Learning the renormalization group itself can open up a window to learning new physics that appears at different scales.

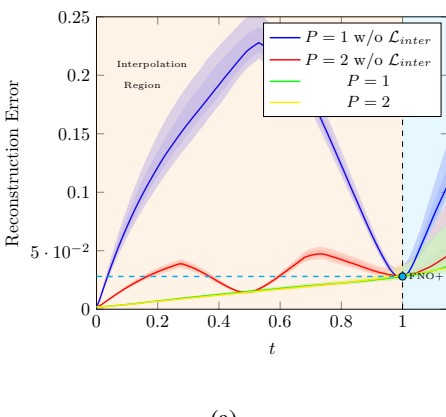 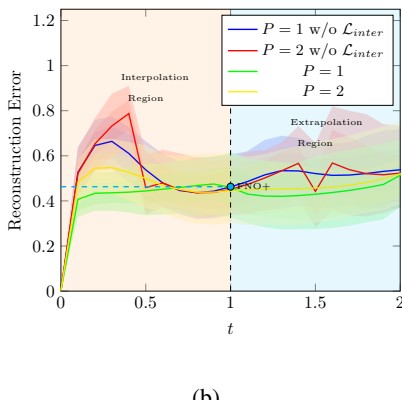

(a)

(b)

Figure 3: The reconstruction error for intermediate time predictions and ground truth solver solutions for different number of intervals $P$ in the composition loss term, $\mathcal{L}_{\text{comp}}^{(P)}$. We show the median as well as the 10th, 25th, 75th and 90th percentiles. The statistics is based on 100 samples on the test set. The dashed horizontal cyan line marks the reconstruction error of FNO at $t = 1$ for comparison. (a) two-dimensional Burgers equation. (b) two-dimensional NS equation.

Table 3: The reconstruction error on the three-dimensional NS equation at $t = 1$.

| Method | 3D NS |
|---|---|
| FNO | 0.2778 |
| FNO+ | 0.2644 |
| Ours w/o $\mathcal{L}_{\text{inter}}$ | 0.2675 |
| Ours+ w/o $\mathcal{L}_{\text{inter}}$ | 0.2672 |
| Ours | **0.2504** |

Fourth, studying the space of solutions and its statistics (in contrast to individual solutions) for PDEs such as NS equations is expected to reveal insights about the universal structure of the physical system. For example, it would be valuable to automatically gain insights into the major unsolved problem of anomalous scaling of turbulence.

In our work we considered a decaying turbulence. In order to reach steady-state turbulence and study the statistics of the fluid velocity distribution one needs to introduce in the network a random force pumping energy to the system at the same rate as that of dissipation. On general ground, one expects that for chaotic systems nearby trajectories would diverge exponentially in time, $\Delta v(t) = \Delta v(0)e^{\lambda t}$. This imposes, in general, a limit on the ability to learn single chaotic solution at long times due to the error accumulation. We note, however, that this is a limitation of any numerical solver.

## 7 CONCLUSIONS

Learning the dynamical evolution of non-linear physical systems governed by PDEs is of much importance, both theoretically and practically. In this work we presented a network scheme that learns the map from initial conditions to solutions of PDEs. The scheme bridges an important gap between data- and model-driven approaches, by allowing data-driven models to provide results for arbitrary times.

Our hyper-network based solver, combined with a Fourier Neural Operator architecture, propagates the initial conditions, while ensuring the general composition properties of the partial differential operators. It improves the learning accuracy at the supervision time-points and interpolates and

extrapolates the solutions to arbitrary (unlabelled) times. We tested our scheme successfully on non-linear PDEs in one, two and three dimensions.

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

## A    Visualization of shocks

Figure 4 shows the time evolution of the states of two-dimensional Burgers equations. Even though only regular solution appeared in the training set, our approach is able to generalize in the vicinity of non-regular solutions as can be seen at $t = 1.15$, where shocks start to develop.

## B    Runtime Analysis

Let $T_{\text{FNO}}$ be the time required for applying FNO once. Both $\mathcal{L}_{\text{initial}}$ and $\mathcal{L}_{\text{final}}$ require a single FNO application, and therefore each is executed in time $T_{FNO}$. The intermediate term, $\mathcal{L}_{\text{inter}}$, requires three FNO calls, and therefore requires $3T_{\text{FNO}}$, and $\mathcal{L}_{\text{Comp}}$ requires $T_{\text{FNO}}$ for each of the partitions, totaling in $5T_{\text{FNO}} + \sum_{p=2}^{P} pT_{\text{FNO}} = 4T_{\text{FNO}} + \frac{1}{2}P(P+1)T_{\text{FNO}}$ calls.

## C    Additional Ablations

To assess the contribution of each term to the loss function in equation 17, different weight coefficients are added,

$$\mathcal{L}_{\text{tot}}^{(P)} = \lambda_{\text{initial}}\mathcal{L}_{\text{initial}} + \lambda_{\text{inter}}\mathcal{L}_{\text{inter}} + \lambda_{\text{supervision}}\mathcal{L}_{\text{supervision}} \cdot \tag{19}$$

where $\mathcal{L}_{\text{supervision}} = \mathcal{L}_{\text{final}} + \mathcal{L}_{\text{comp}}^{(P)}$ (Note that $\mathcal{L}_{\text{final}}$ is the zeroth order of $\mathcal{L}_{\text{comp}}^{(P)}$). The reconstruction error function is computed on the 1D Burgers equation using our method with one partition ($P = 1$). For $\lambda_{\text{supervision}}$ two values are used, $\lambda_{\text{supervision}} = 1$ and $\lambda_{\text{supervision}} = \frac{1}{2}$. For the other weight coefficients we use the values $0.1, 0.5, 1$. All the combinations are summarized in Table 4 for $\lambda_{\text{supervision}} = 1$ and Table 5 for $\lambda_{\text{supervision}} = \frac{1}{2}$. As can be seen, while the precise value of the error varies, the method is robust with respect to these coefficients, except for the case where $\lambda_{\text{initial}}$ is much smaller than both $\lambda_{\text{supervision}}$ and $\lambda_{\text{inter}}$. In this case, it is unable to converge since the initial conditions at $t = 0$ are not learned properly, influencing the hyper-network predictions at $t = 0$ and the whole time evolution.

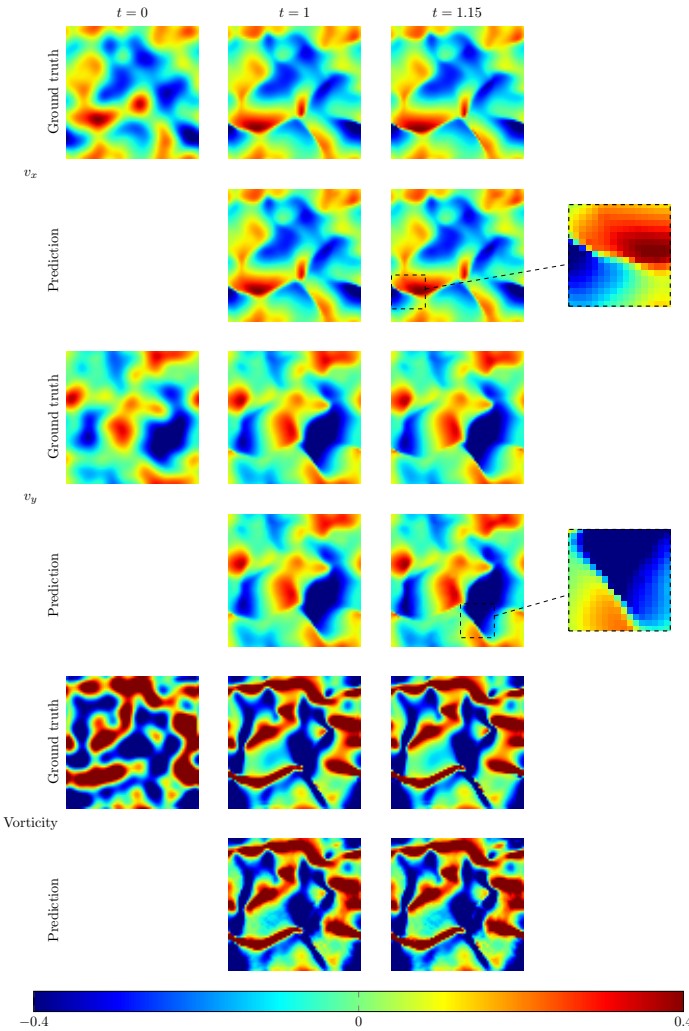

Figure 4: The velocity vector fields and the corresponding vorticity of the two-dimensional Burgers equation. Shocks are being formed in the final state.

Table 4: The reconstruction error on the one-dimensional Burgers equation PDEs at $t = 1$ using a grid size = 1024 and $\lambda_{\text{supervision}} = 1$.

| $\lambda_{\text{initial}}$ | $\lambda_{\text{inter}}$ | Err |
|---|---|---|
| 0.1 | 0.1 | 0.0012 |
| 0.1 | 0.5 | 0.0014 |
| 0.1 | 1.0 | 1.0000 |
| 0.5 | 0.1 | 0.0019 |
| 0.5 | 0.5 | 0.0021 |
| 0.5 | 1.0 | 0.0014 |
| 1.0 | 0.1 | 0.0016 |
| 1.0 | 0.5 | 0.0022 |
| 1.0 | 1.0 | 0.0015 |

Table 5: The reconstruction error on the one-dimensional Burgers equation PDEs at $t = 1$ using a grid size = 1024 and $\lambda_{\text{supervision}} = \frac{1}{2}$.

| $\lambda_{\text{initial}}$ | $\lambda_{\text{inter}}$ | Err |
|---|---|---|
| 0.1 | 0.1 | 0.0023 |
| 0.1 | 0.5 | 0.0019 |
| 0.1 | 1.0 | 0.0019 |
| 0.5 | 0.1 | 0.0024 |
| 0.5 | 0.5 | 0.0014 |
| 0.5 | 1.0 | 0.0013 |
| 1.0 | 0.1 | 0.0021 |
| 1.0 | 0.5 | 0.0020 |
| 1.0 | 1.0 | 0.0018 |

