# OpenReview forum: "Semi-supervised learning of partial differential operators and dynamical flows"
_ICLR.cc/2023/Conference — Submitted to ICLR 2023_

### Official Review · Reviewer_ykCW · 2022-10-20

**Confidence:** 3
**Correctness:** 3
**Technical Novelty And Significance:** 3
**Empirical Novelty And Significance:** 3
**Recommendation:** 3

**Clarity, Quality, Novelty And Reproducibility:**


Clarity adequate, but could be improved. Quality adequate. Novelty adequate. Reproducibility high.


**Strength And Weaknesses:**


s: the result analysis is interesting and quite insightsful.

s: the method and loss seem to be well derived, and are sensible.

s/w: The results are not strong enough. The method does roughly hover around the baseline, getting either marginal improvements or similar results.

w: the writing is ok but could be improved. The introduction could do a better job at describing the problem and the issues in earlier works. This paper is quite incremental by focusing largely on FNO, and making it more powerful. The paper could try to position its contributions on the wider context better.

**Summary Of The Paper:**

This paper presents a hyper-network for FNO’s that allow it to predict intermediate future states. These are used to augment the loss for a better trained model.


**Summary Of The Review:**

This is an ok paper with a sensible idea, but ultimately I feel this is not enough and nothing really stands out. The method is simple and incremental, and the results are not impressive enough.

---

> ### Author Response · Authors · 2022-11-14
> **Author Response**
>
> We thank the reviewer for his helpful comments.
>
> R: “The results are not strong enough. The method does roughly hover around the baseline, getting either marginal improvements or similar results.”
>
> A: Our method improves the reconstruction error on four out of the five 1D benchmarks and on all 2D and 3D benchmarks at the supervision points. The major contribution of our method is that, in contrast to all the other methods including FNO, it provides a solution at arbitrary continuous intermediate timesteps.
>
> R: “the writing is ok but could be improved. The introduction could do a better job at describing the problem and the issues in earlier works. This paper is quite incremental by focusing largely on FNO, and making it more powerful. The paper could try to position its contributions on the wider context better.
> This is an ok paper with a sensible idea, but ultimately I feel this is not enough and nothing really stands out. The method is simple and incremental, and the results are not impressive enough.”
>
> A: In our work we show how to generalize a PDE solving framework that outputs a result at a fixed time T, to continuous intermediate time steps. We chose FNO because it obtains SOTA results for several datasets as our benchmark, but the method is general and can be applied to other solvers using one supervision point. Note that generalizing a discrete time method to a continuous one is not incremental since these are non-linear PDEs and the dynamics is chaotic.

---

> > ### Comment · Reviewer_ykCW · 2022-11-24
> > **response**
> >
> > Thanks for the update. I see the authors' points, but I don't think the work raises above the bar. The method is incremental, simple and a bit heuristic, the presentation is a bit lacking, the results are not impressive with few comparisons and small improvements. I'll keep my score.
> >
> > I haven't mentioned the heuristic criticism before, so I'll expand on it. The results show that the error depends on time granularity, which should not happen. This might be because of the loss, which is approximative, and does not guarantee any constraints. For instance, even initial condition is not guaranteed to be satisfied. Furthermore, the hypernetwork is a bit of a black box, and it's difficult to say what's its effect on this, or what's going on. Overall the method seems to be more engineering-driven than theory-driven.

---

### Official Review · Reviewer_rv2Z · 2022-10-21

**Confidence:** 4
**Correctness:** 2
**Technical Novelty And Significance:** 3
**Empirical Novelty And Significance:** Not applicable
**Recommendation:** 5

**Clarity, Quality, Novelty And Reproducibility:**

The loss seems innovative. The architecture is not new but its application to evolutional PDE solve seems new.

Regarding quality, perhaps it is better to weaken some of the claims. For instance:

* Abstract writes “The evolution of dynamical systems is generically governed by nonlinear partial differential equations”. But it is not “generically”. An example is https://en.wikipedia.org/wiki/Tent_map .
Another example is non-local “PDE”.

* Introduction paragraph 2: “all analytical techniques fail”. This is false and disrespecting an entire field (e.g., [Foias, Manley, Rosa, Temam. Naiver-Stokes equations and turbulence. 2001]).

These are easy to revise though.


**Strength And Weaknesses:**

Strength:
Overall, this is an interesting paper. I especially enjoy the construction of the loss, which prompts what is learned to be like a semigroup.

Weakness:
I wish more empirical results are provided for a more comprehensive evaluation of the performance. I hope I didn’t read incorrectly, but at this moment I have several questions:

(1) Has the method been compared to iterations of vanilla FNO? That is, FNO for characterizing the evolution map over time-step dt, and then iterated T/dt times for reaching time T? I hope the new results still outperform FNO used in this way, for both small dt and large dt.

(2) What kind of Navier-Stokes problem is being considered? What is the initial condition, what should the exact solution look like, and what does the learned solution look like?

(3) In existing methods like NODE, time dependence was often just used as a part of the input of a neural network. Is it possible to justify, experimentally, why the new hierarchical hyper-network structure is better? Is there any performance/computational-cost trade-off?


**Summary Of The Paper:**

Fourier Neural Operator (FNO) is a data-based approach for learning an infinite-dimensional mapping, which is helpful for numerically solving PDEs. The paper extends FNO to an evolutional setup. FNO can be used in an evolutional setup too, for which one can learn a mapping that maps the solution at time t to that at time t+dt, and then iterate this mapping. This paper however aims at directly learning the evolutional semigroup $\Phi(t): \mathcal{H} \to \mathcal{H}$, as a function of both the solution/initial condition and $t$. For this, properties of a semigroup are encouraged by the construction of loss and the time-dependence was accounted for by a specific hyper-network architecture. Interesting results are empirically shown.

**Summary Of The Review:**

Overall I feel this is an interesting paper. Simple but good idea, which can be a solid step toward an important problem (ML for PDE solve). I still have some reservation at this moment due to comments made above about the results, but I can be convinced.

---

> ### Author Response · Authors · 2022-11-14
> **Author Response**
>
> We thank the reviewer for his helpful comments.
>
> R: “Weakness: I wish more empirical results are provided for a more comprehensive evaluation of the performance.”
>
> A:  Our experiments include seven 1D PDEs, two 2D PDEs and an additional 3D PDE where our approach was compared to the current state of the art approach (FNO). Furthermore, we experimented with different hyperparameters (see Appendix B), and different grid sizes (see Figure 1).
>
> R:”Has the method been compared to iterations of vanilla FNO? That is, FNO for characterizing the evolution map over time-step dt, and then iterated T/dt times for reaching time T? I hope the new results still outperform FNO used in this way, for both small dt and large dt.”
>
> A: This approach has been tested at (Li et al. , Learning Dissipative Dynamics in Chaotic Systems, 2022) and has been able to recover the statistical properties of the system, but not the specific trajectories associated with each initial condition. We would also like to emphasize that for each dt, a different FNO model has to be trained (in contrast with our approach).
>
> R:”What kind of Navier-Stokes problem is being considered? What is the initial condition, what should the exact solution look like, and what does the learned solution look like?”
>
> A: The initial conditions for the Navier-Stokes equations are sampled from gaussian random fields using the same parameters as described in the original FNO paper (Li et al. , Fourier neural operator for parametric partial equations, 2019). These parameters appear in the supplementary code. There are no exact analytical solutions to compare with except in the 1d Burgers case.
>
> R: “In existing methods like NODE, time dependence was often just used as a part of the input of a neural network. Is it possible to justify, experimentally, why the new hierarchical hyper-network structure is better? Is there any performance/computational-cost trade-off?”
>
> A:  We assume that the reviewer refers to (Chen et al., Neural Ordinary Differential Equations , 2018). In that work, the equations depend only on one parameter (for instance, time or a spatial coordinate). To apply their approach on a PDE, each point in space and time must be processed individually, and therefore the number of calls to cover a solution grows as the number of space-time dimensions.
>
> R: “Abstract writes “The evolution of dynamical systems is generically governed by nonlinear partial differential equations”. But it is not “generically”. An example is https://en.wikipedia.org/wiki/Tent_map . Another example is non-local “PDE”.”
>
> A: We will correct this and add  “The evolution of many dynamical systems is governed by nonlinear partial differential equations”.
>
> R: “Introduction paragraph 2: “all analytical techniques fail”. This is false and disrespecting an entire field (e.g., [Foias, Manley, Rosa, Temam. Naiver-Stokes equations and turbulence. 2001]).”
>
> A: While as the referee points out and we definitely agree, a lot of effort has been made in order to understand analytically turbulent fluid flows, still the two basic questions have remained unanswered: (1) The existence and uniqueness of the solutions to the 3d Navier-Stokes equations (https://www.claymath.org/millennium-problems/navier%E2%80%93stokes-equation) and (2) The anomalous scaling of the fluid observables in statistical turbulence. We rewrote the sentence in order to reflect these accurately.

---

### Official Review · Reviewer_2qHC · 2022-10-29

**Confidence:** 4
**Correctness:** 4
**Technical Novelty And Significance:** 3
**Empirical Novelty And Significance:** 2
**Recommendation:** 5

**Clarity, Quality, Novelty And Reproducibility:**

The paper is overall understandable, but the clarity can be improved. The novelty is not very impressive.

**Strength And Weaknesses:**

Strength:
- extend operator learning to semisupervised learning
- new hyper-network architecture
- continuous interpolation in time

Weakness:
- learning accuracy at the supervision time-points is not very impressive (vs FNO+)
- need more justification for a semi-supervised learning setting.

Comment:
- Is this model data efficient? Maybe the proposed method will show an advantage at a smaller amount of data sets.
- How does the continuous-time learning compare with the standard autoregressive model (t ->t+1, e.g. Markov neural operator)? The autoregressive automatically enforces composition in time. Since the Navier-Stokes is markovian, one probably does not need history.

**Summary Of The Paper:**

This work considers neural operators in the semisupervised setting. It extends Fourier neural operator with a hypernetwork structure, which allows composition in time. The paper provides numerical experiments on 1d Burger equation and 2d and 3d Navier-Stokes equation. The proposed method has a slightly improved error rate but better interpolation performance in time.

**Summary Of The Review:**

Overall, I think the paper provides a concrete study, but its contribution and novelty may not meet the threshold. Therefore, I think this work is marginally below the bar for acceptance.

---

> ### Author Response · Authors · 2022-11-14
> **Author Response**
>
> We thank the reviewer for his helpful comments.
>
> R: “learning accuracy at the supervision time-points is not very impressive (vs FNO+)”
>
> A: One would expect that our results would match FNO exactly, however, as can be seen, utilizing a hyper-network framework improves (even if by a small margin) the accuracy at the supervision points. However, the main contribution of our method is to be able to interpolate to unseen continuous times.
>
> R: “need more justification for a semi-supervised learning setting.”
>
> A: We chose to call it semi-supervised as there is a supervision at time $t=0$ and $t=1$, while we learn for any time $t \in [0, 1]$ with no direct supervision. This is similar to other semi-supervised settings in which only some of the training samples are labeled. The velocity configurations in the entire range $t \in (0, 1)$ are  “unsupervised samples” and not part of a training set.
>
> R:”Is this model data efficient? Maybe the proposed method will show an advantage at a smaller amount of data sets.”
>
> A: We have tried running our approach with fewer samples, however, the accuracy of our model degraded similarly to FNO and other methods.
>
> R: “How does the continuous-time learning compare with the standard autoregressive model (t ->t+1, e.g. Markov neural operator)? The autoregressive automatically enforces composition in time. Since the Navier-Stokes is markovian, one probably does not need history.”
>
> A: Unlike autoregressive models, which require knowledge of intermediate time steps, our architecture does not require this information. Our approach only utilizes the values of the fields at $t=0$ and $t=1$ (during training), and is able to provide the field values for any intermediate time and for unseen before initial conditions.

---

### Official Review · Reviewer_cCio · 2022-10-31

**Confidence:** 4
**Correctness:** 3
**Technical Novelty And Significance:** 2
**Empirical Novelty And Significance:** 2
**Recommendation:** 3

**Clarity, Quality, Novelty And Reproducibility:**

$\cdot$ This paper is clearly written and organized.

$\cdot$ The novelty of this paper is not obvious as it merely combines known methods. It might be ok to qualify a good novelty as long as there are significant contribution from other aspects, such as empirical results, etc. However, $\mathbf{1}.$ It is not clear how innovative improvement on the architecture of hyper-network; $\mathbf{2}.$ it is not convincing enough the proposed method outperforms other methods as the chose of baseline is limited; $\mathbf{3}.$ predicting intermediate time point is not a novel contribution. Based on the above three points, along with the combo of known methods, the novelty is not obvious.

$\cdot$ The reproducibility is feasible.

**Details Of Ethics Concerns:**

There is no ethics concern in this paper.

**Strength And Weaknesses:**

The strengths of this paper are:

$\mathbf{1}.$ The novelty of separating time and space by the hyper-network architecture.

$\mathbf{2}.$ This paper is clearly written and well organized, with detailed explanation of theories and experiments.

The weakness of this paper are:

$\mathbf{1}.$ The proposed approach is a combination of known methods, although it might be novel to use hyper-network.

$\mathbf{2}.$ The baseline used in the experiments might not be strong enough. For example, why using FNO and/or MGNO in the experiments? Why other network-based methods were not used? Also, is it possible to use some numerical methods as a baseline? The main concern is whether or not the chosen baseline is good and comprehensive to use as references.

Other concerns (minor if compared with the above weakness):

$\mathbf{1}.$ The literature review claims the model base method, such as (Raissi 2019), does not require training dataset. Physics informed network does require training the network with data, as most of the network-based method do.

$\mathbf{2}.$  The literature review also claims that the methods referred do not predict intermediate time, which could be false. Even if it is right, there exists methods which can predict solutions at intermediate time.

$\mathbf{3}.$ The loss function presented in Eq (17) is a simple addition, which could be improved by tuning hyper-parameters as a convex or other types of combinations.

**Summary Of The Paper:**

This paper proposes a hyper-network to solve partial differential equations.

**Summary Of The Review:**

As stated in the above comments, this paper might not deliver a solid contribution, and the novelty could be limited.

---

> ### Author Response · Authors · 2022-11-14
> **Author Response**
>
> We thank the reviewer for his helpful comments.
>
> R: “The baseline used in the experiments might not be strong enough. For example, why using FNO and/or MGNO in the experiments? Why other network-based methods were not used? Also, is it possible to use some numerical methods as a baseline? The main concern is whether or not the chosen baseline is good and comprehensive to use as references.”
>
> A: We used FNO as a baseline since it outperforms the other neural solver approaches by almost an order of magnitude. These methods include:  FCN (Zhu & Zabaras, Bayesian deep convolutional encoder–decoder networks for surrogate modeling and uncertainty quantiﬁcation, 2018), PCANN (Bhattacharya et al. , Model reduction and neural networks for parametric pde(s), 2020), GNO (Li et al.,  Neural operator: Graph kernel network for partial differential equations, 2020) and LNO  (Lu et al., Deeponet: Learning nonlinear operators for identifying differential equations based on the universal approximation theorem of operators, 2019).
>
> R: “The literature review claims the model base method, such as (Raissi 2019), does not require training dataset. Physics informed network does require training the network with data, as most of the network-based method do.”
>
> A: Thank you for pointing out our mistake. Indeed PINNs require training data and we fixed the error in the manuscript. We would like to emphasize that PINNs are not completely data-driven, and require prior knowledge of the PDEs they are intended to solve.
>
> R: “The literature review also claims that the methods referred do not predict intermediate time, which could be false. Even if it is right, there exists methods which can predict solutions at intermediate time.”
>
> A: All the cited works in our manuscript cannot interpolate to continuous intermediate times without any modification to the architecture. To the best of our knowledge no network-based approach has the capability to do so.
>
> R: “The loss function presented in Eq (17) is a simple addition, which could be improved by tuning hyper-parameters as a convex or other types of combinations.”
>
> A: We refer the reviewer to Appendix B in the manuscript that contains an ablation study of importance of different coefficient combinations.
>
> R: “The novelty of this paper is not obvious as it merely combines known methods. It might be ok to qualify a good novelty as long as there are significant contribution from other aspects, such as empirical results, etc. However, It is not clear how innovative improvement on the architecture of hyper-network; it is not convincing enough the proposed method outperforms other methods as the chose of baseline is limited;
> predicting intermediate time point is not a novel contribution. Based on the above three points, along with the combo of known methods, the novelty is not obvious. “
>
> A: We indeed combined two known methods, hyper-networks and FNO, however, these methods alone are not enough, as can be seen in Fig. 2. Only after adding the loss terms in Eq. 17, which is dictated by time-continuity, we are able to both improve the accuracy at the time points that are learned in a supervised manner, and also add the ability to continuously interpolate in time. We also note that on the Navier Stokes 2D, and Burgers 1D datasets, FNO has been shown to surpass other network-based approaches. Furthermore, we would like to point out that the PDEs dealt in our work are non-linear, and therefore, interpolation is highly non trivial, and has not been performed for unseen continuous times by any previous work.

---

### Official Review · Reviewer_NVyD · 2022-11-03

**Confidence:** 3
**Correctness:** 3
**Technical Novelty And Significance:** 3
**Empirical Novelty And Significance:** 3
**Recommendation:** 5

**Clarity, Quality, Novelty And Reproducibility:**

The paper is well-written and easy to digest. As mentioned in the main review, some more information on FNO could be useful.

I haven't had the chance to work with FNO-based approaches yet, thus I cannot evaluate the figures and tables more than what I see in the paper. The presented results look ok to me and the improvement over vanilla FNO in the interpolation regime seems quite clear.

Regarding novelty, the paper seems to introduce the combination of hyper-networks and FNO for the first time. And although the hyper-network and FNO networks seem to be "off the shelf", given the rather specialized loss makes for a novel approach.

The supplementary material seems to do the job in terms of reproducibility. The 1D data is downloadable, and the 2D and 3D data can be generated using PhiFlow. I successfully started a training run on the 1D data.

**Strength And Weaknesses:**

The main strength of the paper is the pursuit to learn the underlying dynamics over the continuous time domain. As the authors mention in the discussion section (Sec. 6), this might indeed be very useful as a reduced-order model e.g. of the NSE.

Comments:
- Given that the main comparison baseline is FNO and FNO is also a major building block in this architecture, I would have expected a brief introduction to the used FNO formalism.
- Where are CNNs and GNNs? Given the recent popularity of GNNs for the solution of PDEs (e.g. https://arxiv.org/abs/2202.03376, https://arxiv.org/abs/2002.09405) as well as CNNs (e.g. https://arxiv.org/abs/1810.08217), one should at least mention these lines of research in the "Related Work" section and justify why they are not (or are) relevant here.
- I see the intention to learn meaningful interpolated solutions of PDEs via the hyper-network, but given the lack of results on extrapolation over longer horizons, e.g. t=10...100, I would speculate that the results are not promising. Any comment on that?  Given the practical relevance of extrapolating over longer times, this aspect should be addressed somewhere. And this regime is something that has already been studied by the so-called direct-time methods, e.g. in this workshop paper by Meyer et al., 2021 using GNNs: https://arxiv.org/abs/2112.10296. I understand that all the PDEs you consider are dissipative, thus too long trajectories wouldn't make sense, but why not look at some inviscid equations (e.g. Burgers, Euler) and discuss e.g. the survival time as done in https://arxiv.org/abs/2202.03376.
- I like the loss terms defined in the paper quite a lot: they incorporate all possible types of interpolation. However, I know other papers that do similar things (ignoring L_final and L_initial for a second): The construction of L_comp^(P) is very closely related to the "temporal bundling" presented here https://arxiv.org/abs/2202.03376, and L_inter reminds me of the "pushforward training" from the same paper. I'm sure that there are more analogies from other papers.

Two minor problems I detected are:
1. some formatting issues with the equations leading to "Equation equation [...]"
2. "Miltiple Graph Neural Operator" -> Multipole Graph Neural Operator"

**Summary Of The Paper:**

The paper presents an approach to modeling PDE solutions using a trainable hyper-network that generates the parameters of an FNO network. The hyper-network gets as an input only the desired continuous time t (>0) and the FNO transforms an initial condition under the dynamics specified by the output of the hyper-network. While the performance of this method at the discrete training time steps already improves on FNO, the main innovation is allowing for interpolated solutions at any continuous time.

**Summary Of The Review:**

The idea of the paper is very interesting, but out of the 21 mentioned references, I didn't find a single one from 2022. This suggests that the authors lack knowledge of the recent literature, which includes e.g. GNNs. This issue and the weak justification of the chosen methods and experiments (see above) are the reason why I believe that this paper is not yet ready for publication.

---

> ### Author Response · Authors · 2022-11-14
> **Author Response**
>
> We thank the reviewer for his helpful comments.
>
> R: “Given that the main comparison baseline is FNO and FNO is also a major building block in this architecture, I would have expected a brief introduction to the used FNO formalism.”
>
> A: Following the reviewer’s suggestion, we’ve added an explanation in the related work section regarding the FNO formalism.
>
> R: “Where are CNNs and GNNs? Given the recent popularity of GNNs for the solution of PDEs (e.g. https://arxiv.org/abs/2202.03376, https://arxiv.org/abs/2002.09405) as well as CNNs (e.g. https://arxiv.org/abs/1810.08217), one should at least mention these lines of research in the "Related Work" section and justify why they are not (or are) relevant here.”
>
> A: In our work we show how to generalize a PDE solving framework which outputs a result at a fixed time T, to continuous intermediate time steps. We chose FNO because it obtains SOTA results for several datasets as our benchmark. Other solvers such as GNNs could also be used, but FNO was a good candidate for its simplicity. We added the references to the other approaches mentioned by the reviewer.
>
> R: “I see the intention to learn meaningful interpolated solutions of PDEs via the hyper-network, but given the lack of results on extrapolation over longer horizons, e.g. t=10...100, I would speculate that the results are not promising. Any comment on that? Given the practical relevance of extrapolating over longer times, this aspect should be addressed somewhere. And this regime is something that has already been studied by the so-called direct-time methods, e.g. in this workshop paper by Meyer et al., 2021 using GNNs: https://arxiv.org/abs/2112.10296. I understand that all the PDEs you consider are dissipative, thus too long trajectories wouldn't make sense, but why not look at some inviscid equations (e.g. Burgers, Euler) and discuss e.g. the survival time as done in https://arxiv.org/abs/2202.03376.”
>
> A: Indeed, it is expected in general that chaotic dynamics makes the extrapolation to late times difficult because of the sensitivity to the initial conditions. However, it is still far from clear that late times dynamics cannot be learnt. The reason that we use dissipation is that turbulence cannot be generated without a small dissipatipative term. While dissipation leads to decaying turbulence, adding a forcing term allows to extrapolate to late times and reach a steady state for the statistics of the solutions.  In order to study turbulent structure using the Euler equation one needs to consider singular solutions, which are harder to learn. When studying the 2D Burgers dynamics, shocks have been developed at early times, which prevented us from extrapolation. In the original FNO paper, where extrapolations to late times was claimed, the forcing term was constant  in time and at late times the solutions reflected the shape of the force.
>
> R: “I like the loss terms defined in the paper quite a lot: they incorporate all possible types of interpolation. However, I know other papers that do similar things (ignoring L_final and L_initial for a second): The construction of L_comp^(P) is very closely related to the "temporal bundling" presented here https://arxiv.org/abs/2202.03376, and L_inter reminds me of the "pushforward training" from the same paper. I'm sure that there are more analogies from other papers.”
>
> A:  The issue with temporal bundling trick such as presented in the https://arxiv.org/abs/2202.03376 is that in order for the approach to obtain small-time resolution scales, it is required to execute the method with an increasing number of sequential model compositions  (see Algorithm 1 in Appendix F in https://arxiv.org/abs/2202.03376). For instance, to obtain a $\frac{1}{10}$ ratio, one would need to run the FNO network $10$ times, thus the time resolution scales with the network’s depth, and each time scale requires a different architecture. The construction in our approach has no limitation on the minimal scale, and allows the prediction of intermediate time steps with fine tuning to different scales, without any additional computational resources.
>
> R: “Two minor problems I detected are:
> some formatting issues with the equations leading to "Equation equation [...]"
> "Miltiple Graph Neural Operator" -> Multipole Graph Neural Operator" “
>
> A: We will correct these typos. Thank you.
>
> R: “The idea of the paper is very interesting, but out of the 21 mentioned references, I didn't find a single one from 2022. This suggests that the authors lack knowledge of the recent literature, which includes e.g. GNNs. This issue and the weak justification of the chosen methods and experiments (see above) are the reason why I believe that this paper is not yet ready for publication.”
>
> A: We have been following the literature and will be happy to refer to papers that are directly related to our work and that we might have missed. We added the references suggested by the reviewer.

---

> > ### Comment · Reviewer_NVyD · 2022-12-09
> > **response**
> >
> > Many thanks for incorporating some of my comments in the paper!
> >
> > However, my concern regarding the limitation of the proposed approach for extrapolation in time, in combination with the still weak justification of the choice of the method compared to state-of-the-art alternatives, results in keeping my recommendation unchanged.

---

### Decision · Program_Chairs · 2023-01-20

**Decision:**

Reject

**Justification For Why Not Higher Score:**

Marginal innovation and experimental validation insufficient

**Justification For Why Not Lower Score:**

N/A

**Metareview: Summary, Strengths And Weaknesses:**

The paper introduces an extension of Fourier Neural Operator (FNO) for solving PDEs and forecasting dynamical flows. The novelty consists in using a hypernetwork for modeling the weight evolution of FNO inside the forecasting horizon. Compared to the baseline FNO, supervision is added at intermediate time steps in the forecasting interval for training and the loss function is enriched with additional terms. Evaluation is performed on a series of simulated PDEs in 1 and 2 dimensions.

The main innovation w.r.t. FNO is the possibility to predict in continuous time via the hypernetwork. All the reviewers consider that the novelty is limited and that the evaluation should be enriched with additional baselines. The paper is interesting but too preliminary for acceptance.